# α-Linolenic Acid Inhibits RANKL-Induced Osteoclastogenesis In Vitro and Prevents Inflammation In Vivo

**DOI:** 10.3390/foods12030682

**Published:** 2023-02-03

**Authors:** Yufeng Deng, Weizhou Li, Yingying Zhang, Jingjing Li, Fangting He, Ke Dong, Zehui Hong, Ruocheng Luo, Xiaofang Pei

**Affiliations:** 1West China School of Public Health, West China Fourth Hospital, Sichuan University, Chengdu 610041, China; 2Department of Healthcare-Associated Infection Management, The Affiliated Hospital of Guizhou Medical University, Guiyang 550004, China; 3Department of Laboratory Medicine, Chengdu Second People’s Hospital, Chengdu 610041, China; 4Food Safety Monitoring and Risk Assessment Key Laboratory of Sichuan Province, Department of Public Health Laboratory Sciences, West China School of Public Health, Sichuan University, Chengdu 610041, China; 5Non-Communicable Diseases Research Center, West China-PUMC C. C. Chen Institute of Health, Sichuan University, Chengdu 610041, China

**Keywords:** alpha-linolenic acid, osteoclastogenesis, anti-inflammation, ovariectomized rat, *Zanthoxylum bungeanum* seed oil

## Abstract

Inflammation is an important risk factor for bone-destroying diseases. Our preliminary research found that *Zanthoxylum bungeanum* seed oil (ZBSO) is abundant in unsaturated fatty acids and could inhibit osteoclastogenesis in receptor activator of nuclear factor κB ligand (RANKL)-induced RAW264.7 cells. However, the key constituents in ZBSO in the prevention of osteoclastogenesis and its possible mechanism related to inflammation are still unclear. Therefore, in this study, oleic acid (OA), linoleic acid (LA), palmitoleic acid (PLA), and alpha-linolenic acid (ALA) in ZBSO, havingthe strongest effect on RANKL-induced osteoclastogenesis, were selected by a tartrate-resistant acid phosphatase (TRAP) staining method. Furthermore, the effects of the selected fatty acids on anti-inflammation and anti-osteoclastogenesis in vitro and in vivo were assessed using RT-qPCR. Among the four major unsaturated fatty acids we tested, ALA displayed the strongest inhibitory effect on osteoclastogenesis. The increased expression of free fatty acid receptor 4 (FFAR4) and β-arrestin2 (βarr2), as well as the decreased expression of nuclear factor-kappa B (NF-κB), tumor necrosis factor-α (TNF-α), nuclear factor of activated T-cells c1 (NFATc1), and tartrate-resistant acid phosphatase (TRAP) in RAW264.7 cells after ALA treatment were observed. Moreover, in ovariectomized osteoporotic rats with ALA preventive intervention, we found that the expression of TNF-α, interleukin-6 (IL-6), interleukin-1β (IL-1β), NFATc1, and TRAP were decreased, while with the ALA therapeutic intervention, downregulated expression of NF-κB, NFATc1, TRAP, and transforming growth factor beta-activated kinase 1 (TAK1) were noticed. These results indicate that ALA, as the major unsaturated fatty acid in ZBSO, could inhibit RANKL-induced osteoclastogenesis via the FFAR4/βarr2 signaling pathway and could prevent inflammation, suggesting that ZBSO may be a promising potential natural product of unsaturated fatty acids and a dietary supplement for the prevention of osteoclastogenesis and inflammatory diseases.

## 1. Introduction

Although inflammation is a defense and repair response for dealing with damage or infection of tissues and organs [1], it may result in local or systemic pathological diseases such as colitis [2], atherosclerosis [3] and osteoporosis [4], due to the over-expression of inflammatory factors. Khosla et al. [5] reported that the occurrence and development of postmenopausal osteoporosis was closely related to the decline of female estrogen and chronic inflammation. Inflammatory factors, such as IL-6, IL-1β and TNF-α, may alter bone cell generation through the NF-κB signaling pathway [6,7], MAPK signaling pathway [8,9], Wnt/β-catenin signaling pathway [10,11] and other signaling pathways, tilting the original bone metabolism balance toward bone resorption and leading to bone destruction and bone loss [12]. Moreover, the increased risk of fracture in the presence of local or systemic bone loss in some patients with inflammatory diseases, such as rheumatic inflammatory bowel disease and systemic sclerosis, are reported [13,14]. These studies suggest that inflammation may act on osteoclasts, affect bone health, disturb the balance of bone metabolism, and result in adverse effects on bone remodeling. Therefore, the prevention and treatment of bone loss via inhibiting inflammation could be a promising strategy. Notably, investigations into natural products that have anti-inflammatory activity with lower costs and fewer side effects have been increasing in recent years [15,16].

*Zanthoxyhum bungeanum* Maxim (ZBM), a plant of the *Rutaceae* family, is mostly planted in Sichuan, Gansu and Shanxi provinces of China [17]. It is a traditional spice and a medicinal cash crop in China, and can be used to treat asthma [18], tumors [19], thrombus [20], etc. *Zanthoxylum bungeanum* seed oil (ZBSO) is a vegetable oil extracted from *Zanthoxylum bungeanum* Maxim seed, the anti-inflammatory and antioxidant effects of which have been recently reported [21,22]. ZBSO is rich in unsaturated fatty acids [23] at approximately 70% of its content (*w*/*w*), and mainly includes oleic acid (OA), linoleic acid (LA), palmitic acid (PLA), alpha-linolenic acid (ALA), etc. [18,23]. The ability of ZBSO to inhibit RANKL-induced osteoclastogenesis has been observed in our previous research [24]. Moreover, the regulation of bone metabolism by unsaturated fatty acids through alleviating inflammation, inhibiting apoptosis, and regulating oxidative stress have been reported [25]. Hence we speculate that unsaturated fatty acids such as ALA, OA, and LA in ZBSO may be the major constituents for the inhibition of osteoclastogenesis.

Therefore, we conducted a comparison study on the inhibitory effects of the major unsaturated fatty acids (LA, OA, PLA, ALA) in ZBSO on RANKL-induced osteoclastogenesis using the TRAP staining method. Subsequently, the potential mechanisms of the selected fatty acids on modulating inflammatory factors and RANKL-induced osteoclastogenesis were assessed with RT-qPCR. Furthermore, ovariectomized osteoporotic rats were used to study the preventive and therapeutic effects of the selected unsaturated fatty acids on inflammatory osteoporosis.

## 2. Materials and Methods

### 2.1. Cell Culture and Cell Induction

RAW264.7 cells (Stem Cell Bank, Chinese Academy of Sciences, Shanghai, China) were cultured and passaged in the DMEM (supplemented with 100 units/mL penicillin, 100 μg/mL streptomycin and 10% FBS), while osteoclastogenic medium (OCM) containing DMEM and 50 ng/mL RANKL (Absin Bioscience Inc, Shanghai, China) was used for differentiation induction. The cells treated with DMEM, OCM, and OCM containing 0.1% BSA and 0.05% ethanol, were defined as the control group (Control), the model control group (Model), and the vehicle control group (Vehicle), respectively, and incubated at 37 °C with 5% CO_2_ humidification.

### 2.2. Preparation of Fatty Acids

Four kinds of fatty acids were purchased from ANPEL Laboratory Technologies (Shanghai, China) Inc. In accordance with previous reports [26,27], 50.8 mg PLA (purity: 99%), 56.1 mg LA (purity: 99%), 56.5 mg OA (purity: 98.9%), and 55.6 mg ALA (purity: 99%) was dissolved in 500 μL anhydrous ethanol and 9.5 mL 10% BSA, heating at 55 °C, respectively, to prepare the fatty acids stock solution with the concentration of 20 mM, filtered with a 0.22 μm filter and stored at −20 °C. The original solutions were diluted with OCM to the different concentrations required.

### 2.3. Cell Viability Assays

RAW264.7 cells were seeded in 96-well plates at a density of 4 × 10^5^ per well and cultured to a cell density of about 85%. Referring to previous research [28,29], different concentrations of PLA, OA, LA and ALA (25 µM, 50 µM, 75 µM, 100 µM, 125 µM, 150 µM, 175 µM and 200 µM) were added into DMEM and incubated at 37 °C and 5% CO_2_ for 24 h, 100 μL/well, respectively. The DMEM was used as a cell control group, and 0.1% BSA with 0.05% ethanol was used as the vehicle control group. After 24 h, we added 10% CCK-8 reagent (Japan Dojin Institute of Chemistry), 100 μL/well, cultured at 37 °C and 5% CO_2_ for 2 h. The absorbance values were determined by a microplate reader at 450 nm (Thermo Fisher Scientific, Waltham, MA, USA).

### 2.4. TRAP Staining

RAW264.7 cells were incubated at a concentration of 1 × 10^4^ per well and cultured in DMEM at 37 °C, 5% CO_2_ for 24 h. After 24 h, cells were washed with PBS, and OCM was added into the plate, and then PLA, LA, OA and ALA (50 μM, 100 μM, 200 μM) were added to the OCM at 37 °C, 5% CO_2_, 100 μL/well, with each concentration being set with three multiple wells. Meanwhile, the Control, Model, and Vehicle groups were set up. Cells were cultured for 120 h, and the medium was replenished at 48 h and 96 h. After 120 h, cells were immobilized with 4% paraformaldehyde for 20 min and stained for TRAP (Sigma-Aldrich, St. Louis, MO, USA), as explained. TRAP positive cells with >3 nuclei were counted by Image Pro Plus software using a light microscope (Leica Cameral AG, Wetzlar, Germany) and were counted as osteoblasts.

### 2.5. Animals and Experimental Design

Eighty female Sprague-Dawley rats (aged eight weeks) obtained from Beijing Vital River Laboratory Animal Technology Co., Ltd. (animal production license: Beijing Baishan SCXK (Beijing, China) 2021-0006) were kept in a room equipped with controlled humidity (50–60%) and temperature (23 ± 2 °C), with 12/12 h light and dark cycles and adaptive feeding for one week.As shown in Figure 1, after acclimatization, the rats were assigned into eight groups, ten rats for each, and anesthetized with pentobarbital (40 mg/kg/mL). Sixty rats with bilateral ovariectomization were used as a model for osteoporosis, while the remaining twenty rats were subjected to the same operation but their ovaries were not removed. The sixty rats with bilateral ovariectomization were further divided into six groups, and were treated with preventive and therapeutic intervention, respectively. Preventive intervention was administered 1 week after ovariectomy or adipose tissue removal, while therapeutic intervention was administered 12 weeks later. The rats were subjected to different interventions for 12 weeks before they were sacrificed. The two interventions were grouped in the same way as follows: unresected ovary (SHAM, intragastric administration of 5 mL/kg bacteria-free saline every day) as the sham group, ovariectomized (OVX, intragastric administration of 5 mL/kg bacteria-free saline every day) as the model control group, ovariectomized and given estradiol (OVX + E2, intragastric injections of 5 mL/kg estradiol every day) as the positive control group, and ovariectomized and treated with ALA (OVX + ALA, intragastric administration of 1.09 mL/kg ALA every day) as the intervention group. The dosage of ALA was according to the content of ALA in ZBSO.

During intervention, a weekly measurement of the rat’s body weight was taken, and the dosage was adjusted accordingly. After 12 h of fasting on the final day of the intervention, the rats were tranquilized by injecting 3% pentobarbital solution intraperitoneally and blood was harvested from the abdominal aorta, clarified by centrifugation at 3000 rpm for 10 min to extract serum, and stored at −80°C for subsequent analysis. After complete excision of muscle and connective tissues, femur and tibia were also harvested for assessment.

For all animal experimental procedures, we followed the Laboratory Animal and Protection Regulations, and the experimental scheme was authorized by the Ethics Committee of West China Fourth Hospital and West China School of Public Health, Sichuan University (Batch no. Gwll2021054).

### 2.6. Micro-CT

Micro-CT (SCANCO Medical AG) was applied to scan the right femur of rats, with three rats in each group being randomly selected for detection of their bone mineral density. The right femur was placed in the Micro-CT scanner and scanned at 360° with the following parameters: voltage 70 kV, current 114 μA, and resolution 10 μm.

### 2.7. Blood Routine Examination and Serum Inflammatory Factor Analysis

Automatic blood cell analyzer (Beckman Coulter Co., LTD, Brea, CA, USA) was used to detect routine blood indexes of rats, including white blood cells (WBC), neutrophilic granulocytes (NE), lymphocytes (LY), monocytes (MO), red blood cells (RBC), hemoglobin (HGB), and platelets (PLT). Neutrophil-to-lymphocyte ratio (NLR) and platelet-to-lymphocyte ratio (PLR) were also calculated. Double antibody sandwich ELISA (Wuhan Elite Biotechnology Co., Ltd., Wuhan, China) was used for the determination of serum TNF-α, IL-6 and IL-1β expression levels, according to the kit instructions.

### 2.8. RNA Isolation and Real-Time Quantitative 

PCRRAW264.7 cells were inoculated at a concentration of 4 ×10^5^ per well and cultured in DMEM at 37 °C and 5% CO_2_ for 24 h, followed by treatment of the different concentrations of ALA (50 μM, 100 μM, 200 μM) in OCM for 24 h in the same culture conditions. After centrifugation, the cells were collected. RNA-easy Isolation Reagen was used to extract total RNA from each group of cells, and Trizol reagent was used to extract total RNA from each group of tibiae. Then, 2 μg of RNA was extracted for cDNA synthesis using the iScript cDNA synthesis kit.RT-qPCR was performed using the SYBR^®^ Green Premix Pro Taq HS qPCR kit and detected using the CFX96 Touch Real-Time PCR Detection System (BIO-RAD, Hercules, CA, USA) (95 °C for 30 s, 95 °C for 5 s, 60 °C for 5 s, for 40 cycles). β-actin were used as the internal standard to determine the relative values of RAW264.7 cells mRNA expression (the primer sequences are presented in detail in Table 1), while housekeeping gene Gapdh of rat cells were used as the internal standard (the primer sequences are presented in detail in Table 2). The results were obtained by the 2^−ΔΔCt^ method.

### 2.9. Statistical Analyses

Experimental data are represented as mean ± SD. GraphPad Prism 7.0 (GraphPad Software, La Jolla, CA, USA) was utilized for all statistical analyses. One-way analysis of variance (ANOVA) was performed to analyze significant differences between multiple comparisons. A difference of *p* < 0.05 is regarded as statistically significant.

## 3. Results

### 3.1. Fatty Acids Inhibited RANKL-Induced Osteoclastogenesis

RAW264.7 cells were processed with four various concentrations of fatty acids for 24 h (50–200 μM), and, compared to the control and vehicle groups, the cell viability rate of the RAW264.7 cells was not significantly decreased. Fatty acids of 50 μM, 100 μM and 200 μM were used for further tests. The cell viability rate among all groups is illustrated in Figure 2.

No significant inhibitory effects of PLA and OA on RANKL-induced osteoclastogenesis were observed (*p* > 0.05) (Figure 3a,e,c,g). The inhibitory effect of LA was noticed only at a high concentration (200 µM) (*p* < 0.01). On the contrary, for ALA, the inhibitory effects were observed in all concentrations tested (*p* < 0.001) (Figure 3d,h). These outcomes suggest that ALA has the strongest effect on inhibiting osteoclastogenesis among the four major fatty acids, thus ALA was chosen for subsequent experiments.

### 3.2. ALA Inhibited RANKL-Induced Osteoclastogenesis Genes Expression

RT-qPCR was applied to evaluate the expression of specific osteoclastogenesis genes after the treatment of ALA. The expression of FFAR4 and βarr2 was upregulated (*p* < 0.01, *p* < 0.05, respectively), while NF-κB expression was downregulated (*p* < 0.001) in the ALA-H (200 µM) group as compared to the model group; moreover, the expression of these genes was dose-dependent after ALA treatment. In addition, the expected trend of TAK1 expression was also observed (Figure 4a–d). The gene expression of TNF-α, NFATc1, and TRAP treated with RANKL was dramatically enhanced (*p* < 0.05), however, these could be downregulated by ALA in a dose-dependent manner (*p* < 0.05) (Figure 4e,g,h). However, a slightly decreased expression of IL-6 was observed after the treatment of ALA, although this difference was not statistically significant (*p* > 0.05) (Figure 4f).

### 3.3. Effects of ALA on Body Weight and BMD

After three months preventive intervention, the body weight of the rats in all groups except the SHAM group was significantly increased (Figure 5a), and the bone mineral density (BMD) in the OVX group was decreased compared with the SHAM group (101.6 ± 2.67 vs. 361.3 ± 28.35, *p* < 0.001), which implied that the rat models of osteoporosis were successfully developed. Interestingly, a decreased body weight was observed in the ALA group compared with the OVX and E2 groups (114.4 ± 7.52 vs. 169.8 ± 6.43, 182 ± 9.99, *p* < 0.01) (Figure 5a). BMD was higher in the E2 group than in the OVX group (182 ± 9.99 vs. 101.6 ± 2.67, *p* < 0.05). However, the difference in BMD between the ALA and OVX groups was not significant (85.24 ± 9.14 vs. 101.6 ± 2.67, *p* > 0.05) (Figure 5b), which implied that the effect of ALA on improving BMD was not obvious.After three months of therapeutic intervention, the body weight gain of the OVX group was slightly higher than that of the SHAM group, although this difference was not statistically significant (43.13 ± 8.32 vs. 29.45 ± 5.35, *p* > 0.05). However, the body weight gain of the ALA group was significantly lower than those of the SHAM and OVX groups (1.72 ± 2.37 vs. 29.45 ± 5.35, 43.13 ± 8.32, *p* < 0.05). As expected, ovariectomy produced a significant increase in the weight of the rats, and ALA intake reverted this change (Figure 5c). The change of BMD in therapeutic intervention is compatible with the change in preventive intervention (Figure 5d).

### 3.4. Effects of ALA on Routine Blood Indexes

As shown in Table 3, with the preventive intervention, the differences in WBC, NE, LY, MO, PLT, and PLR numbers were not statistically significant (*p* > 0.05), and NLR was increased (*p* < 0.05) in the OVX group compared with the SHAM group. Compared with the OVX group, WBC and LY numbers were increased (*p* < 0.05), and RBC, NLR and PLR numbers were decreased (*p* < 0.05) in the ALA group.

With the therapeutic intervention, there was an increase in the number of RBC, HGB and PLR in the OVX group compared to the SHAM group (*p* < 0.05). Comparing the ALA group with the OVX group, the number of WBC and LY were increased (*p* < 0.05), while the number of PLR was decreased (*p* < 0.05) (Table 4).

### 3.5. Effects of ALA on Serum Inflammatory Factors In Vivo

The expression levels of TNF-α, IL-6 and IL-1β were increased with preventive intervention in the OVX group compared to the SHAM group (*p* < 0.05). Compared with the OVX group, the expression levels of TNF-α and IL-6 in the E2 and the ALA groups were decreased (*p* < 0.05), while the expression level of IL-1β was not decreased significantly (*p* > 0.05) (Figure 6a–c). However, in the therapeutic intervention groups, no significant differences were observed related to these inflammatory factors (*p* > 0.05) (Figure 7a–c).

### 3.6. Effects of ALA on Expression of Inflammatory Factors and Osteoclast Genes in Bone of Rats

In preventive intervention groups, the expression of TNF-α, IL-6, IL-1β, NFATc1, TRAP, TAK1 and NF-κB in the bone of the OVX group were upregulated as compared to the SHAM group (*p* < 0.0001, *p* < 0.001, *p* < 0.05, *p* < 0.0001, *p* < 0.0001, *p* < 0.01, *p* < 0.01, respectively), suggesting that the inflammation had occurred in the bone of the model rats. After treatment with ALA, we found that the expression of IL-6, NFATc1, TRAP, TAK1 and NF-κB were decreased significantly (*p* < 0.001, *p* < 0.05, *p* < 0.001, *p* < 0.0001, *p* < 0.05, respectively) as compared with those of the OVX group, indicating that ALA could alleviate the expression of inflammatory factors and osteoclast genes. Although significant differences in the expression of TNF-α, IL-1β, and βarr2 were not observed as compared with those of the OVX group, their expected trends were noticed (Figure 8).

In therapeutic intervention groups, a significant upregulated expression of IL-6 and TRAP were observed in the bone of the OVX group compared to the SHAM group (*p* < 0.0001, *p* < 0.001, respectively), and the expected trends of the expression of TNF-α, IL-1β, NFATc1, FFAR4, TAK1 and NF-κB were noticed as well. After the treatment with ALA, a significant decreased expression of IL-1β, NFATc1, TRAP and TAK1 (*p* < 0.0001, *p* < 0.0001, *p* < 0.01, *p* < 0.0001, respectively), and the expected expression trends of IL-6, NF-κB, FFAR4 and βarr2 were also observed by comparison with those of the OVX group (Figure 9), suggesting that ALA could alleviate the inflammation of ovariectomized rats and may influence osteoclastogenesis.

## 4. Discussion

In this present study, we first conducted a comparison evaluation on the inhibitory effects of the four major unsaturated fatty acids in ZBSO on osteoclastogenesis with TRAP staining, and we found that the strongest anti-osteoclastogenesis effect was demonstrated after treatment with ALA. Subsequently, we found that the effect of ALA on anti-osteoclastogenesis may occur through regulating the FFAR4/βarr2 signaling pathway. Moreover, animal experiments were further applied and we observed that both preventive and therapeutic intervention with ALA could decrease the inflammation in ovariectomized osteoporotic rats, although the effects with preventive intervention were stronger.

This is the first reported comparison study of the four major fatty acids in ZBSO related to their inhibitory effect on osteoclastogenesis. After treatment with OA, LA, PLA and ALA, respectively, the strongest inhibitory effect was displayed with the ALA intervention (Figure 3). ALA is an essential nutrient for human health, which cannot be produced by the body itself and must be provided in the diet [46]. The crucial roles of ALA in anti-inflammation, cardiovascular disease prevention, and bone metabolism regulation, as well as its anti-inflammatory effect on human corneal epithelial cell mediated through NF-κB signal transduction, are reported [47]. In our study, the concentrations of ALA tested were 50 μM, 100 μM, and 200 μM. After treatment for 24 h, a decreased number of osteoclasts induced by RANKL (Figure 3d,h) were observed in all concentrations of ALA tested, with the effective concentrations being lower than those reported in Song’s study [48].

Although ALA could inhibit osteoclastogenesis and prevent inflammation via downregulation of NF-κB-iNOS signaling pathways, as reported [48], other possible pathways by which ALA inhibits osteoclast differentiation have scarcely been investigated. Oh et al. [49] reported that DHA and EPA could bind to FFAR4, activate βarr2, restrain TAK1, and negatively regulate the expression NF-κB. FFAR4, as the key component of the FFAR4/βarr2 signaling pathway acting as one of the fatty acid receptors [50], mainly expressed in macrophages, adipocytes, gastrointestinal endocrine cells, etc., could be activated by unsaturated fatty acids [51] and could further influence the expression of NFATc1. NFATc1, an important transcription factor regulating osteoclastogenesis, could promote the expression of osteoclast genes such as *MMP*-9 and *TRAP* [52]. Therefore, we speculated that the depressive effect of ALA on the differentiation of osteoblasts may also be related to its ability to activate the FFAR4/βarr2 signaling pathway, and we confirmed our hypothesis in this study. We found that the effect of ALA on anti-osteoclastogenesis might be through activating the FFAR4/βarr2 signaling pathway by reducing the expression of downstream *NF*-κB gene and further downregulating the expression of NFATc1, TRAP, TNF-α and IL-6 (Figure 4).

After in vitro experiments revealed the possible mechanisms of ALA on anti-osteoclastogenesis and anti-inflammation, we further established the ovariectomized rat model and treated the rats with ALA with both preventive and therapeutic intervention, in order to evaluate its inhibitory effects on osteoporosis and inflammation in vivo. ALA could inhibit the body weight gain of ovariectomized rats with both preventive and therapeutic interventions, which is consistent with studies on ALA as a dietary supplement for losing weight [53,54]. The findings show that ALA did not improve BMD in osteoporotic rats, indicating that the dosage of ALA in ZBSO cannot improve osteoporosis in vivo. Moreover, we found that the dosage of ALA used in this study was different from other researchers. Polat et al. [55] and Fu et al. [56] found that ALA significantly increased BMD levels at doses of 25, 50 and 200 mg/kg. At the time this study, there was no recommended human intake of ZBSO, so the intake of ZBSO was referenced to peony seed oil, 6 g per day [57]. The dosage of ALA used (1.09 mL/kg) was based on the ZBSO intake and on body surface area conversion between human and laboratory animals. It was 30 times the intake of ALA content in ZBSO. ALA could inhibit the differentiation of osteoclasts in vitro, however, the same results were not obtained in vivo.

The number of lymphocytes was increased with the treatment of ALA (Table 3 and Table 4), suggesting that ALA was likely to activate immune active cells in vivo and had potential anti-inflammation activity, which was in agreement with our findings in vitro. Furthermore, similar research regarding the increased number of lymphocytes after ALA treatment has also been reported [58]. The dysregulated lymphocytes could initiate a cascade mediated by inflammatory factors and chemotactic cytokines, induce neutrophils and macrophages to aggregate, and enhance the bone resorption function of osteoclasts [59,60]. ALA also could inhibit the increase of NLR and PLR caused by ovariectomy with preventive intervention (Table 3), indicating its effects on anti-inflammation and anti-osteoclastogenesis in vivo. NLR and PLR were used as systemic indicators to evaluate the state of systemic inflammation and immune response [61], and their relationships with BMD reduction and anti-osteoporosis has also been reported in recent studies [62,63]. These studies partially supported our findings.

Furthermore, decreases of TNF-α and IL-6 in the serum of rats were observed in preventive intervention but not in therapeutic intervention (Figure 6 and Figure 7), indicating that ALA might be preferable in use for the prevention of inflammation. TNF-α and IL-6 are closely related to the function of osteoclasts, and the high expression of inflammatory factors is a risk factor for osteoporosis [64]. Although we found that ALA could inhibit osteoclastogenesis via suppressing the activation of the FFAR4/βarr2 signaling pathway in RAW264.7 cells, similar results were not obtained in ovariectomized rat models (Figure 8 and Figure 9), which calls for further study.

## 5. Conclusions

To sum up, we demonstrated for the first time that the in vitro inhibitory effect of ZBSO on RANKL-induced osteoclastogenesis and its anti-inflammatory role may be contributed to ALA regulation of the FFAR4/βarr2 signaling pathway. Therefore, we hypothesize that ZBSO could be a prospective natural product of unsaturated fatty acids and a dietary supplement for the prevention of osteoclastogenesis and inflammatory diseases.

## Figures and Tables

**Figure 1 foods-12-00682-f001:**
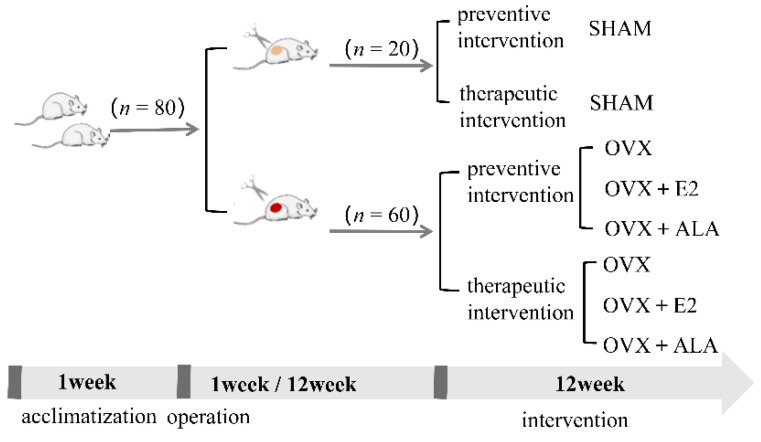
Rats grouping and intervention methods.

**Figure 2 foods-12-00682-f002:**
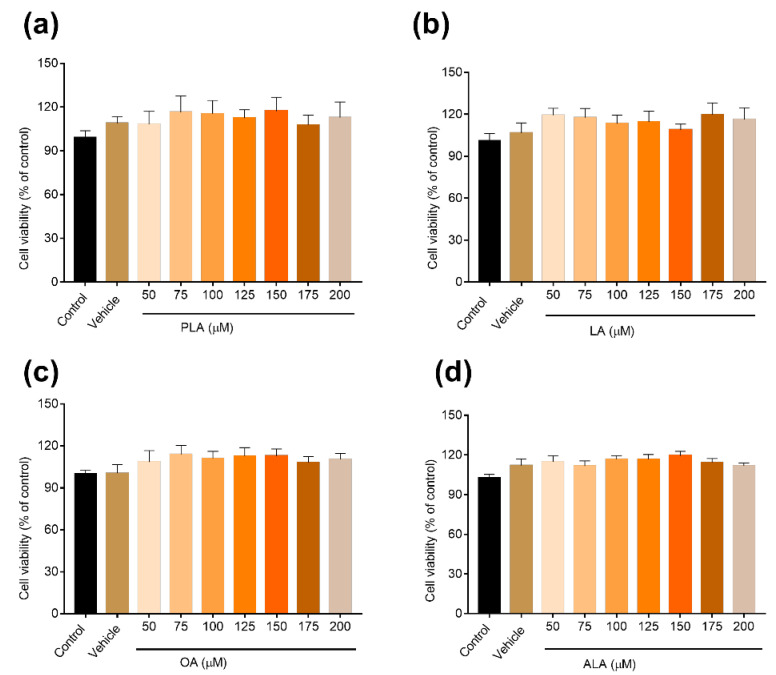
Effects of fatty acids on the cell viability percentage of RAW264.7 cells. (**a**) PLA, (**b**) LA, (**c**) OA, (**d**) ALA.

**Figure 3 foods-12-00682-f003:**
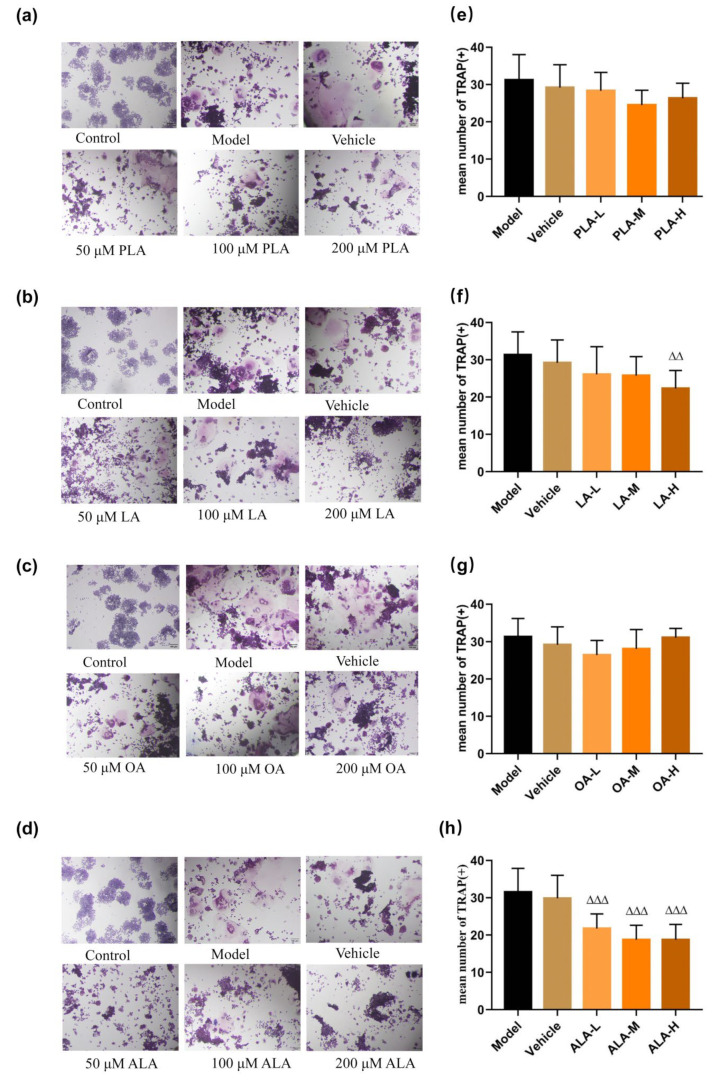
Inhibitory effects of fatty acids on RANKL-induced osteoclastogenesis. (**a**–**d**) Results of TRAP staining with optical microscope. (**e**–**h**) TRAP positive cells with >3 nuclei were counted by optical microscope (×100). (**a**,**e**) PLA, (**b**,**f**) OA, (**c**,**g**) LA, (**d**,**h**) ALA, ^ΔΔ^*p* < 0.01, ^ΔΔΔ^*p* < 0.001 vs. Model (RANKL (+)).

**Figure 4 foods-12-00682-f004:**
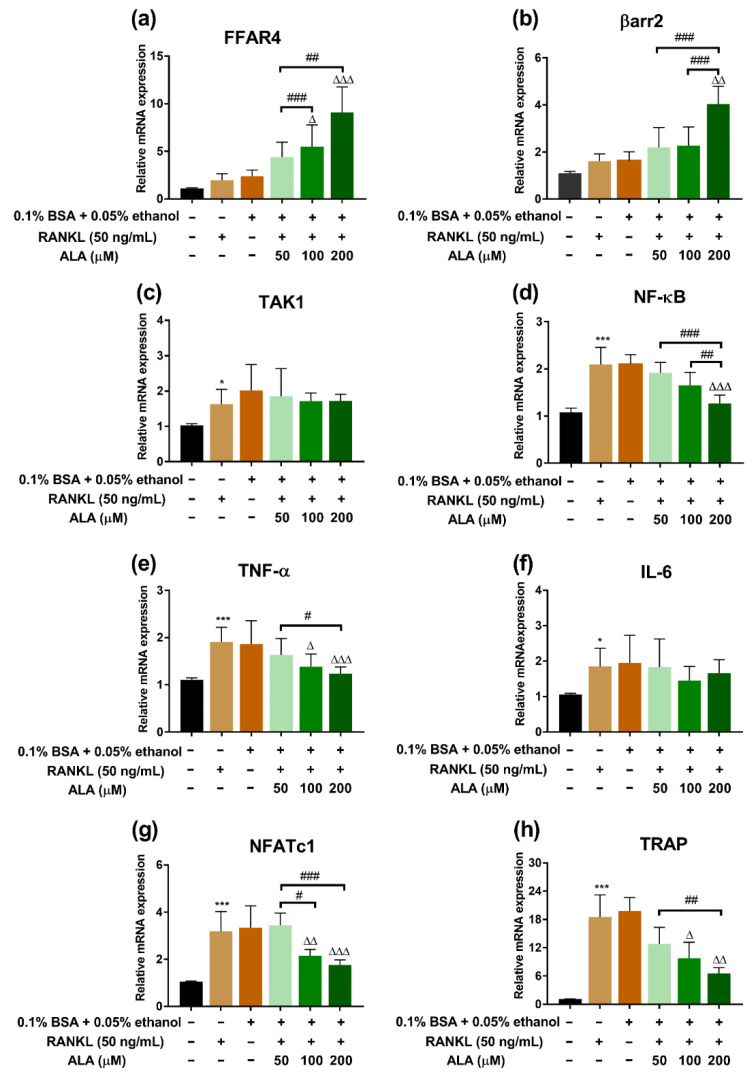
Effects of ALA on FFAR4/βarr2 signaling pathway in vitro. The mRNA expression of (**a**) FFAR4, (**b**) βarr2, (**c**) TAK1, (**d**) NF-κB, (**e**) TNF-α, (**f**) IL-6, (**g**) NFATc1 and (**h**) TRAP in RANKL-induced RAW264.7 cells with or without ALA treatment were measured by RT-qPCR. * *p* < 0.05, *** *p* < 0.001 vs. Control (0.1% BSA + 0.05% ethanol (−), RANKL (−), ALA (−)). ^Δ^
*p* < 0.05, ^ΔΔ^
*p* < 0.01, ^ΔΔΔ^
*p* < 0.001 vs. Model (0.1% BSA + 0.05% ethanol (−), RANKL (+), ALA (−)). ^#^
*p* < 0.05, ^##^
*p* < 0.01, ^###^
*p* < 0.001 represents the comparison between the two groups. + and − indicate with and without treatment, respectively.

**Figure 5 foods-12-00682-f005:**
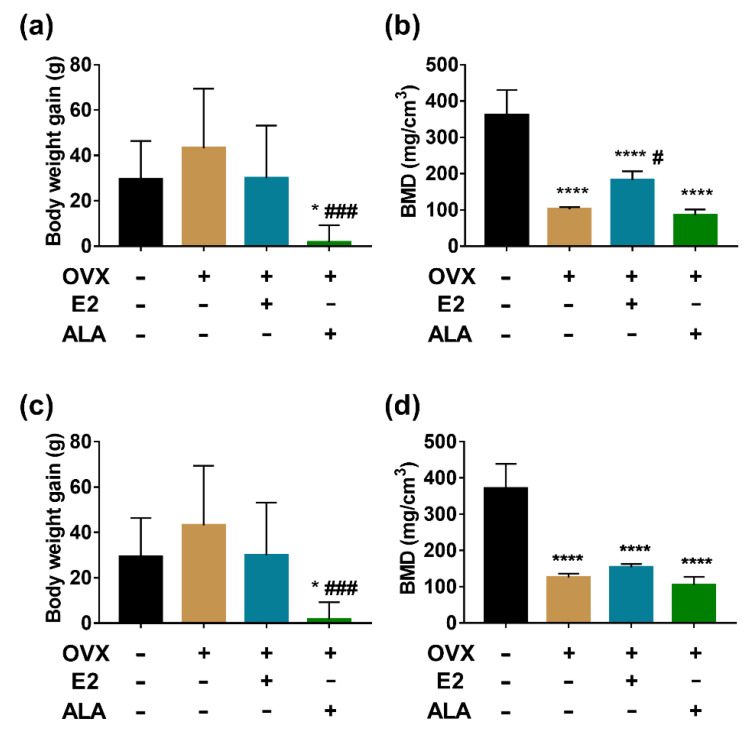
Effects of ALA on body weight gain and BMD in vivo. The changes in (**a**) body weight gain and (**b**) BMD of ovariectomized rat model with or without ALA preventive intervention. The changes in (**c**) body weight gain and (**d**) BMD of ovariectomized rat model with or without ALA therapeutic intervention. * *p* < 0.05, ^****^
*p* < 0.0001 vs. the SHAM group (OVX (−), E2 (−), ALA (−)). ^#^
*p* < 0.05 ^###^
*p* < 0.001, vs. the OVX group (OVX (+), E2 (−), ALA (−)).

**Figure 6 foods-12-00682-f006:**
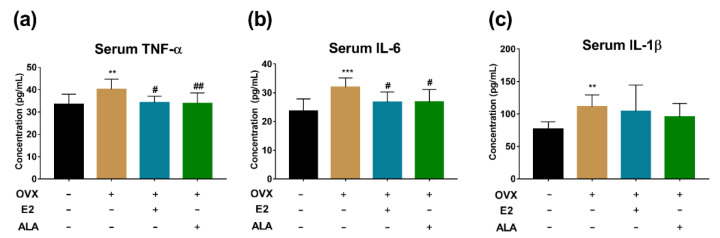
Effects of ALA preventive intervention on the expression levels of serum inflammatory factors. The expression levels of (**a**) TNF-α, (**b**) IL-6 and (**c**) IL-1β in serum of ovariectomized rat models with or without ALA preventive intervention were measured by ELISA. ** *p* < 0.01, *** *p* < 0.001 vs. the SHAM group (OVX (−), E2 (−), ALA (−)). ^#^
*p* < 0.05, ^##^
*p* < 0.01 vs. the OVX group (OVX (+), E2 (−), ALA (−)).

**Figure 7 foods-12-00682-f007:**
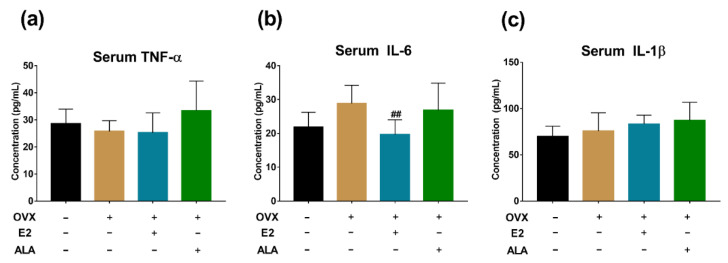
Effects of ALA therapeutic intervention on the expression levels of serum inflammatory factors. The expression levels of (**a**) TNF-α, (**b**) IL-6 and (**c**) IL-1β in serum of ovariectomized rat models with or without ALA therapeutic intervention were measured by ELISA. ^##^
*p* < 0.01 vs. the OVX group (OVX (+), E2 (−), ALA (−)).

**Figure 8 foods-12-00682-f008:**
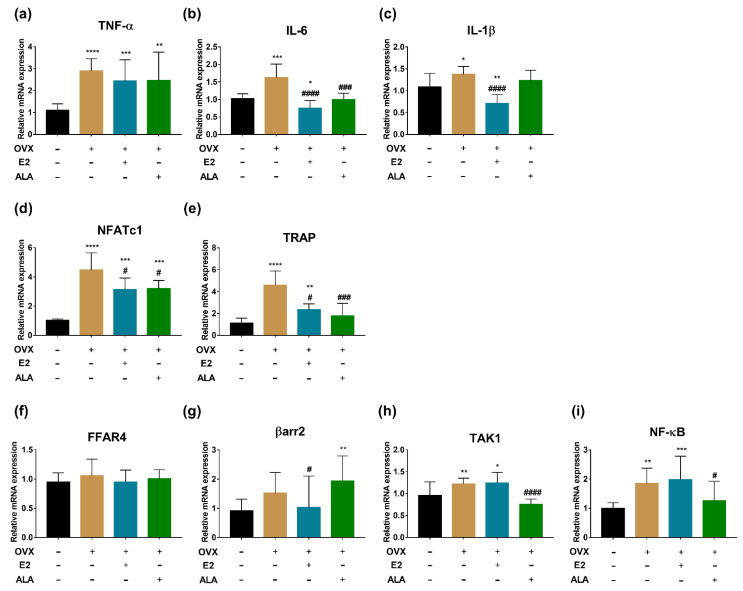
The expression of inflammatory factors and osteoclast genes in the bone of ovariectomized rats with ALA preventive treatment. (**a**) TNF-α, (**b**) IL-6, (**c**) IL-1β, (**d**) NFATc1, (**e**) TRAP, (**f**) FFAR4, (**g**) βarr2, (**h**) TAK1 and (**i**) NF-κB expression were measured by RT-qPCR. * *p* < 0.05, ** *p* < 0.01, *** *p* < 0.001, ^****^
*p* < 0.0001 vs. the SHAM group (OVX (−), E2 (−), ALA (−)). ^#^
*p* < 0.05, ^###^
*p* < 0.001, ^####^
*p* < 0.0001 vs. the OVX group (OVX (+), E2 (−), ALA (−)).

**Figure 9 foods-12-00682-f009:**
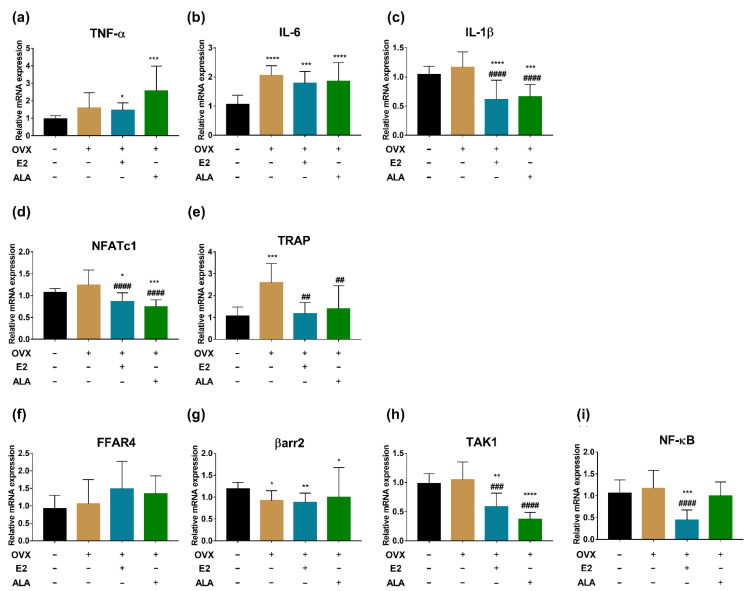
The expression of inflammatory factors and osteoclast genes in the bone of ovariectomized rats with ALA therapeutic treatment. (**a**) TNF-α, (**b**) IL-6, (**c**) IL-1β, (**d**) NFATc1, (**e**) TRAP, (**f**) FFAR4, (**g**) βarr2, (**h**) TAK1 and (**i**) NF-κB expression were measured by RT-qPCR. * *p* < 0.05, ** *p* < 0.01, *** *p* < 0.001, ^****^
*p* < 0.0001 vs. the SHAM group (OVX (−), E2 (−), ALA (−)). ^##^
*p* < 0.01, ^###^
*p* < 0.001, ^####^
*p* < 0.0001 vs. the OVX group (OVX (+), E2 (−), ALA (−)).

**Table 1 foods-12-00682-t001:** Primer sequences of the genes for RAW264.7 cells in RT-qPCR.

Gene	Primer Sequences (5′→3′)	Reference
*FFAR4*	Forward: CTGGGGCTCATCTTTGTCGT	[30]
	Reverse: ACGACGAGCACTAGAGGGAT	
*βarr2*	Forward: ATCACTTGTTGAAAGTGGGC	[31]
	Reverse: GTCTCGTCTTCAAGGATTGG	
*TAK1*	Forward: CCTCCTCGTCTTCTGCCAGTGA	[32]
	Reverse: ACTCCAAAAGCTCCTCTTCCGACA	
*NFATc1*	Forward: CGTTGCTTCCAGAAAATAACA	[33]
	Reverse: TGTGGGATGTGAACTCGGAA	
*TRAP*	Forward: CTGGGGCTCATCTTTGTCGT	[34]
	Reverse: CCCCAGAGACATGATGAAGTCA	
*IL-6*	Forward: CCACTTCACAAGTCGGAGGCTTA	[35]
	Reverse: GCAAGTGCATCATCGTTGTTCATAC	
*TNF-α*	Forward: ATGAGAAGTTCCCAAATGGC	[36]
	Reverse: CTCCACTTGGTGGTTTGCTA	
*NF-κB*	Forward: ATGGCAGACGATGATCCCTAC	[37]
	Reverse: TGTTGACAGTGGTATTTCTGGTG	
*β* *-actin*	Forward: TCTGCTGGAAGGTGGACAGT	[38]
	Reverse: CCTCTATGCCAACACAGTGC	

**Table 2 foods-12-00682-t002:** Primer sequences of the genes for rats in RT-qPCR.

Gene	Primer Sequences (5′→3′)	Reference
*Gapdh*	Forward: AGTGCCAGCCTCGTCTCATA	[39]
	Reverse: TGAACTTGCCGTGGGTAGAG	
*TNF-α*	Forward: AACTCGAGTGACAAGCCCGTAG	[40]
	Reverse: GTACCACCAGTTGGTTGTCTTTGA	
*IL-6*	Forward: ACCCCAACTTCCAATGCTC	[39]
	Reverse: GGTTTGCCGAGTAGACCTC	
*IL-1β*	Forward: CCTTGTCGAGAATGGGCAGT	[41]
	Reverse: ACCAGAATGTGCCACGGTTT	
*FFAR4*	Forward: CCACCGTTCTGGGACTCATC	[39]
	Reverse: CTCCACTTGGTGGTTTGCTA	
*βarr2*	Forward: TGGGCAACTCAAGCACGA	[42]
	Reverse: AGCTTCACCTTGACCCTGTAGGA	
*TAK1*	Forward: AGCAGAAACGACAAGGCACT	[43]
	Reverse: CAGCGAGACAGTGGATTTGA	
*NF-κB*	Forward: GAGACCTGGAGCAAGCCATT	[44]
	Reverse: CAGGCTAGGGTCAGCGTATG	
*NFATc1*	Forward: TGGAGAAGCAGAGCACAGAC	[45]
	Reverse: GCGGAAAGGTGGTATCTCAA	
*TRAP*	Forward: GCTGGAAACCATGATCACCT	[45]
	Reverse: GAGTTGCCACACAGCATCAC	

**Table 3 foods-12-00682-t003:** Routine blood indexes in different rat groups with preventive intervention.

Detection Indexes	SHAM	OVX	E2	ALA
WBC (10^9^ per/mL)	3.11 ± 0.57	4.61 ± 1.5	4.13 ± 1.59	8.14 ± 1.63 *
NE (10^9^ per/mL)	0.66 ± 0.32	1.14 ± 0.55	0.89 ± 0.54	1.03 ± 0.65
LY (10^9^ per/mL)	3.11 ± 0.91	2.6 ± 0.98	4.4 ± 1.71	6.85 ± 1.23 ^#^
MO (10^9^ per/mL)	0.26 ± 0.09	0.33 ± 0.17	0.35 ± 0.11	0.41 ± 0.16
RBC (10^9^ per/mL)	7.40 ± 0.30	8.66 ± 0.28 *	8.31 ± 0.37	8.17 ± 0.27 ^#^
HGB (g/L)	150.25 ± 6.2	170.88 ± 5.25 ^#^	163.13 ± 4.16	162.00 ± 6.28
PLT (10^9^ per/mL)	689.38 ± 54.79	648 ± 76.81	707.13 ± 90.95	647.38 ± 88.91
NLR	0.21 ± 0.09	0.45 ± 0.21 *	0.21 ± 0.09 ^#^	0.15 ± 0.09 ^#^
PLR	235.44 ± 56.62	285.85 ± 89.37	183.92 ± 70.64 ^#^	96.76 ± 18.95 ^*,#^

** p* < 0.05 vs. the SHAM group. # *p* < 0.05 vs. the OVX group.

**Table 4 foods-12-00682-t004:** Routine blood indexes in different rat groups with therapeutic intervention.

Detection Indexes	SHAM	OVX	E2	ALA
WBC (10^9^ per/mL)	2.36 ± 1.01	1.45 ± 0.36	1.70 ± 0.41	3.04 ± 1.10 ^#^
NE (10^9^ per/mL)	0.25 ± 0.13	0.30 ± 0.18	0.30 ± 0.19	0.73 ± 0.30
LY (10^9^ per/mL)	1.74 ± 0.26	1.13 ± 0.39	1.4 ± 0.42	2.55 ± 1.01 ^#^
MO (10^9^ per/mL)	0.36 ± 0.11	0.29 ± 0.16	0.2 ± 0.11	0.31 ± 0.10
RBC (10^9^ per/mL)	7.02 ± 0.13	7.57 ± 0.44 *	7.54 ± 0.19	7.52 ± 0.70
HGB (g/L)	137.38 ± 3.34	146.13 ± 9.51 *	145.75 ± 5.73	147.13 ± 8.46
PLT (10^9^ per/mL)	899.38 ± 177.32	789.13 ± 63.73	757.25 ± 251.28	782.13 ± 148.11
NLR	0.17 ± 0.09	0.29 ± 0.18 *	0.24 ± 0.13	0.30 ± 0.14
PLR	524.46 ± 107.96	768.39 ± 262.63 *	552.06 ± 205.7 ^#^	343.34 ± 119.37 ^#^

* *p* < 0.05 vs. the SHAM group. # *p* < 0.05 vs. the OVX group.

## Data Availability

The data are available from the corresponding author.

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
