# Peer review of "α-Linolenic Acid Inhibits RANKL-Induced Osteoclastogenesis In Vitro and Prevents Inflammation In Vivo"

_foods, 2023, doi:10.3390/foods12030682_

Round 1

Reviewer 1 Report

1) Abstract.In conclusion, ALA, as the major unsaturated fatty acid in ZBSO, exhibited the strongest inhibitory effect on osteoclastogenesis, could prevent inflammation in ovariectomized osteoporotic rats, suggesting that ZBSO may serve as a promising nature resource for prevention of osteoporosis. Please, improve the description of conclusions and underline the novelty of the study.

2) Introduction.L53-55. Inflammatory factors such as IL-6, IL-1β  and TNF-α could promote osteoclastogenesis via affecting RANKL-related signaling  pathways, tilting the original bone metabolism balance toward bone resorption, resulting in bone destruction and bone loss [6]  In order to discuss the previously reported points, important references are needed to be added, such as: 

Dickkopf-1 (Dkk-1) serum levels in systemic sclerosis and rheumatoid arthritis patients: correlation with the Trabecular Bone Score (TBS). Clin Rheumatol. 2018;37(11):3057-3062. doi:10.1007/s10067-018-4322-9

Bone Metabolism Alterations in Systemic Sclerosis: An Insight into Bone Disease in SSc: From the Radiographic Findings to their Potential Pathogenesis and Outcome. Curr Rheumatol Rev. 2022;18(4):286-297. doi:10.2174/1573397118666220218112703

Vitamin D deficiency and clinical correlations in systemic sclerosis patients: A retrospective analysis for possible future developments. PLoS One. 2017;12(6):e0179062. Published 2017 Jun 9. doi:10.1371/journal.pone.0179062

3) Introduction. L81-83. Furthermore, the effects of selected unsaturated fatty acid on  ovariectomized model rats treated with preventive and therapeutic intervention were  studied. Please, improve the description of study aim and underline the novelty of the paper.

4) 3. Results. 3.1 Fatty acids inhibited RANKL-induced osteoclastogenesis. Please, underline in the manuscript the most important statistical data to support the results.

5) 4. Discussion L334-336 Although our previous study found that ZBSO could inhibit osteoclastogenesis in 335 RAW264.7 cells induced by RANKL [17], the key constituents on prevention of  osteoclastogenesis and its possible mechanism related to inflammation were still unclear. Please, summarise here the most important results of the current study.

6) 5. Conclusion L412-419. In summary, this is the first study to report the key constituents in ZBSO on  prevention of osteoclastogenesis and its possible mechanism related to inflammation. We found that ALA had the strongest effect on anti-osteoclastogenesis among four major  unsaturated fatty acids in ZBSO. We also found that ALA could activate FFAR4, βarr2  signaling molecules, leading to the reduced expression of NF-κB, NFATc1, TRAP, TNF417 α and IL-6 gene. Besides, our results also demonstrated that ALA had a better anti inflammatory effect with the preventive intervention in ovariectomized rats. Therefore,  we speculate that ZBSO might be a promising natural resource of unsaturated fatty acids  for the prevention of osteoclastogenesis and inflammation. Please, improve the conclusions and underline the novelty of the study and the possible clinical implications.

Reviewer 2 Report

Deng and colleagues demonstrate the potential use of α-linolenic acid against RANKL-induced osteoclastogenesis in vitro and in vivo, but several issues should be addressed.

1.      In the abstract, the author should provide the complete form of all abbreviations as it is used as the first time.   

2.      Section 2.2 Author should provide the purity of all purchased fatty acids.

3.      Line 118: what is the meaning of OCM?

4.      Figure 4 is too unclear to understand. Authors should provide a more precise and prominent figure for better understanding.  For example: the name of x and y bar as well as the representative name of each figure, is not clear in the present form.

5.      Figure 5 also need modification to clearly understand the group name and x and y bar factor.

6.      Figures 6, 7, and 8 also need clarification. 
